# Tifcemalimab as monotherapy or in combination with toripalimab in patients with relapsed/refractory lymphoma: a Phase I trial

Yuqin Song[1,16], Jun Ma[2,16], Huilai Zhang[3], Yan Xie[1], Zhigang Peng[4], Yuerong Shuang[5], Fei Li[6], Yufu Li[7], Haiyan Yang[8], Liqun Zou[9], Xiuhua Sun[10], Weili Zhao[11], Wenrong Huang[12], Yunhong Huang[13], Hui Zhou[14], Yifan Wang[15], Weiwei Wang[15], Jing Xu[15], Rong Deng[15], Qin Meng[15] & Jun Zhu [1] ✉

Preclinical studies of tifcemalimab (anti-BTLA antibody) in combination with toripalimab (anti-PD-1 antibody) demonstrated synergistic anti-tumor effects. We present the outcomes of tifcemalimab with or without toripalimab in lymphoma patients. This is a 2-part, phase I study (NCT04477772). In Part A (dose escalation based on 3 + 3 design), patients with relapsed or refractory lymphoma received tifcemalimab monotherapy 1, 3 or 10 mg/kg for dose escalation and 3 mg/kg or 200 mg for dose expansion. For Part B (indication expansion), only classical Hodgkin's lymphoma (cHL) patients were included to receive tifcemalimab 100 or 200 mg plus toripalimab 240 mg due to poor tumor response in other subtypes. The primary endpoints were safety, maximum tolerated dose (MTD) and recommended phase 2 dose (RP2D). 25 patients in Part A and 46 in Part B were enrolled. No dose-limiting toxicities were observed, and MTD was not reached. The RP2D for tifcemalimab was 200 mg. Adverse events were predominantly Grade 1/2. Grade 3/4 treatment-related adverse events occurred in 3 patients (12·0%) in Part A and 15 patients (32·6%) in Part B. No fatal adverse events were observed. Tifcemalimab with or without toripalimab demonstrated a favorable safety profile in lymphoma patients.

Programmed cell death (PD)-1 monoclonal antibody, such as pembrolizumab, nivolumab, tislelizumab, sintilimab, and camrelizumab, is a standard treatment for relapsed or refractory (R/R) classical Hodgkin's lymphoma (cHL)[1,2]. However, about 30% of the patients will develop primary resistance to PD-(L)1 blockade, and the majority of the rest may relapse within 1 to 2 years[2]. Subsequent treatments options for those patients are extremely limited. In addition, benefits of PD-(L)1 blockade in patients with non-Hodgkin's lymphoma are

[1]Peking University Cancer Hospital & Institute, Beijing, China. [2]Harbin Institute of Hematology & Oncology, Harbin, China. [3]Tianjin Medical University Cancer Institute and Hospital, Tianjin, China. [4]The First Affiliated Hospital of Guangxi Medical University, Nanning, China. [5]Jiangxi Cancer Hospital, Nanchang, China. [6]The First Affiliated Hospital of Nanchang University, Nanchang, China. [7]Henan Cancer Hospital, Zhengzhou, China. [8]Zhejiang Cancer Hospital, Hangzhou, China. [9]West China Hospital of Sichuan University, Chengdu, China. [10]The Second Hospital of Dalian Medical University, Dalian, China. [11]Shanghai Ruijin Hospital, Shanghai Jiaotong University School of Medicine, Shanghai, China. [12]The Fifth Medical Center of the General Hospital of the Chinese People's Liberation Army, Beijing, China. [13]The Affiliated Cancer Hospital of Guizhou Medical University, Guiyang, China. [14]Hunan Cancer Hospital, Changsha, China. [15]Shanghai Junshi Biosciences, Shanghai, China. [16]These authors contributed equally: Yuqin Song, Jun Ma. ✉e-mail: zhu-jun2017@outlook.com

modest. Thereby, effective therapies to address resistance to PD-(L)1 blockade in advanced lymphoma are in demand in clinic.

B and T lymphocyte attenuator (BTLA), a member of the CD28 receptor family, is expressed on T and B lymphocytes and dendritic cells (DCs)[3]. It is an inhibitory receptor involved in negative regulation of T cell function through the interaction with its ligand herpes virus entry mediator (HVEM)[3,4], which is expressed on a wide range of cells, such as T cells, B cells, natural killer cells, myeloid cells, DCs and tumor cells (e.g., lymphoma, non-small cell lung cancer, melanoma, and colorectal cancer)[5–7]. Mechanically, the combination of PD-1 and BTLA blockade shows a synergistic effect as PD-1 selectively recruits SHP2 over the stronger phosphatase SHP1[8], while BTLA preferentially recruits SHP1 to more efficiently suppress T cell signaling[9]. It has been reported that there is a higher increase in melanoma-specific cytotoxic T lymphocytes (CTL) counts and production of effector cytokines when treated with dual BTLA and PD-1 blockade compared to BTLA or PD-1 blockade alone[9]. Preclinical studies of tifcemalimab (anti-BTLA antibody) and toripalimab (anti-PD-1 antibody) also showed that BTLA blockade works synergistically along with PD-1 blockade to restore T cell function in reporter cells with high BTLA and PD-1 expression and to enhance anti-tumor effect in mouse models. Therefore, simultaneously blocking BTLA-HVEM and PD-1/PD-L1 pathways may produce benefits to cancer treatment.

Tifcemalimab (JS004/TAB004) is the first-in-class anti-BTLA IgG4 antibody under clinical development. Toripalimab is an anti-PD-1 Ig4κ antibody, which has been approved for the treatment of multiple malignancies in China and the United States. We conducted a multicenter, multicohort, phase 1 study (NCT04477772) to explore the safety and preliminary efficacy of tifcemalimab with or without toripalimab in patients with relapsed or refractory lymphoma.

In this work, we report the main findings of the study, which demonstrate a manageable safety profile of that tifcemalimab monotherapy or in combination with toripalimab in lymphoma patients. Furthermore, the combination therapy shows promising clinical efficacy in cHL patients, with an ORR of 37.0%, and a median PFS of 13.1 months (95% CI: 6.4, 16.4).

## Results
### Patients
Between July 17, 2020, and January 29, 2023, 71 patients with lymphoma were enrolled (25 patients in Part A and 46 patients in Part B) from 14 cancer hospitals or general hospitals in China. Limited anti-tumor activity was observed in patients with peripheral T-cell lymphoma, follicular lymphoma, and diffuse large B-cell lymphoma, which led to enrollment early termination for patients with those subtypes. Preliminary anti-tumor activities were observed in cHL patients, therefore, the study continued to enroll cHL patients as planned. Totally, 71 patients were enrolled as of the study completion. In Part A, 9 patients were enrolled for dose escalation, and 16 were enrolled for dose expansion in mixed lymphoma subtypes. The indication-expansion phase was suspended due to insufficient antitumor activity. In Part B, 6, 14, and 26 patients (all patients were cHL) were enrolled for dose escalation, dose expansion, and indication expansion, respectively. The indication expansion for other subtypes was halted due to poor tumor response. At the study completion (February 08, 2024), all the patients in Part A and 34 patients (34/46, 73.9%) in Part B discontinued the study treatments, and the most common reason was disease progression (22/25 in Part A and 21/46 in Part B) (Fig. 1). The median overall survival (OS) follow up was 81.3 weeks (range: 3.0–174.1).

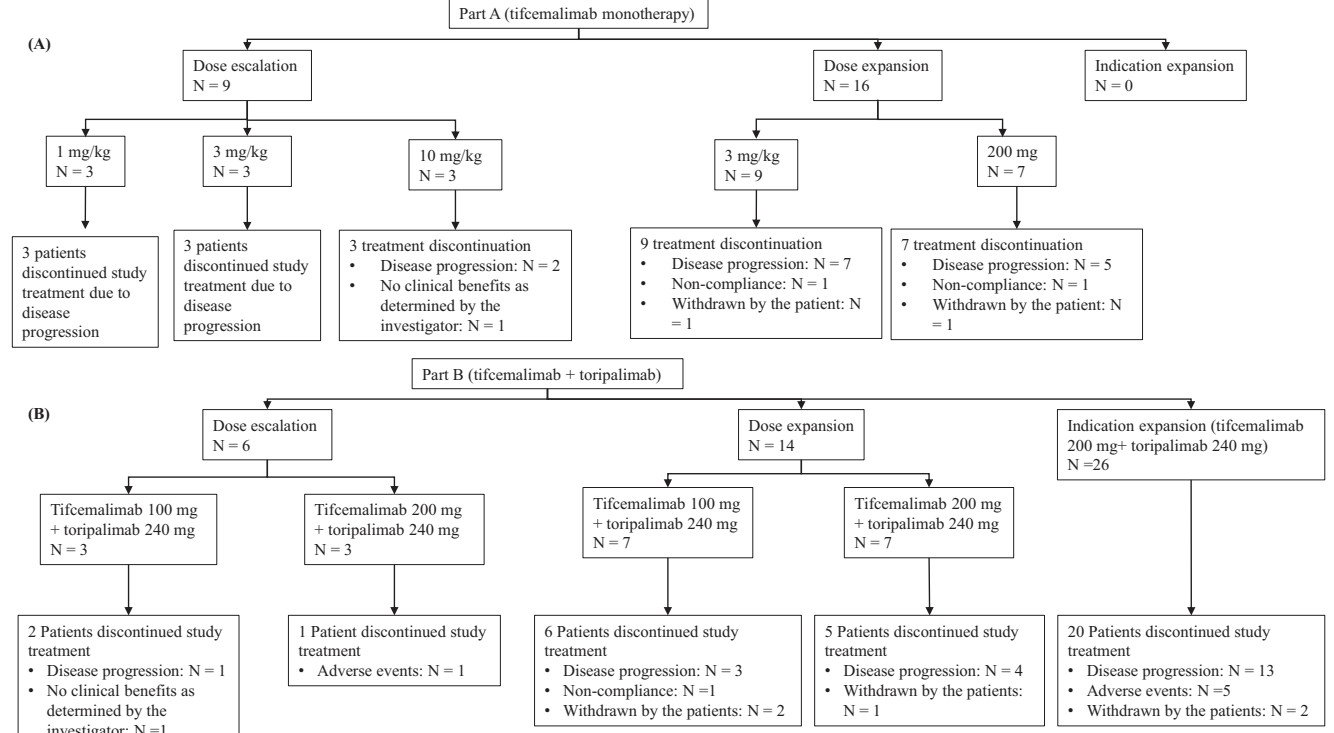

**Fig. 1 | Patient Disposition for Part A and Part B.** As of study completion (February 28, 2024), all patients in Part A (Figure **A**) and 34 patients in Part B (Figure **B**) discontinued the study treatment, and 12 patients are still on treatment due to clinical benefits. In Part B, 6 patients (all in the tifcemalimab 200 mg cohort) developed adverse events leading to study drug discontinuation, including 1 patient with Grade 4 anaphylactic shock, 1 patient with Grade 3 pneumonitis, 1 patient with Grade 4 sepsis, 1 patient with Grade 3 immune-mediated hepatitis, 1 patient with Grade 3 abnormal hepatic function, and 1 patient with Grade 3 drug eruption. Source data are provided as a Source Data file for Fig. 1. N means the number of patients.

**Table 1 | Demographic and Baseline Characteristics (Full Analysis Set)**

| | Monotherapy (Part A) N = 25 | Combination therapy (Pat B) N = 46 |
|---|---|---|
| Age (years) | | |
| Median (min-max) | 45 (26–70) | 36 (19–68) |
| < 60, n (%) | 19 (76.0) | 43 (93.5) |
| ≥ 60, n (%) | 6 (24.0) | 3 (6.5) |
| Sex[a], n (%) | | |
| Female | 11 (44.0) | 10 (21.7) |
| Male | 14 (56.0) | 36 (78.3) |
| ECOG PS, n (%) | | |
| 0 | 21 (84.0) | 26 (56.5) |
| 1 | 4 (16.0) | 20 (43.5) |
| B symptom, n (%) | | |
| Yes | 1 (4.0) | 9 (19.6) |
| No | 24 (96.0) | 37 (80.4) |
| Ann-Arbor stage n (%) | | |
| I/II | 1 (4.0) | 8 (17.4) |
| III/IV | 24 (96.0) | 38 (82.6) |
| LDH expression, n (%) | | |
| LDH ≤ 240 | 14 (56.0) | 29 (63.0) |
| LDH>240 | 11 (44.0) | 17 (37.0) |
| Prior PD-(L)1 treatment, n (%) | | |
| Yes | 13 (52.0) | 41 (89.1) |
| No | 12 (48.0) | 5 (10.9) |
| Prior treatment line | | |
| Median (min-max) | 4 (1-9) | 3 (1–12) |
| Lymphoma subtypes, n (%) | | |
| Follicular lymphoma | 8 (32.0) | 1 (2.2) |
| Classical Hodgkin lymphoma | 9 (36.0) | 42 (91.3) |
| Peripheral T-cell lymphoma | 2 (8.0) | 3 (6.5) |
| Diffuse large B-cell lymphoma | 6 (24.0) | 0 |
| Prior treatment for Classical Hodgkin lymphoma | | |
| ASCT | 2 (22.2) | 11 (26.2) |
| BV | 0 | 5 (11.9) |
| Anti-CD30 ADC investigation | 4 (44.4) | 6 (14.3) |
| PD-(L)1 blockade | 9 (100) | 41 (97.6) |
| PD-1 | 8 (88.9) | 38 (90.5) |
| PD-L1 | 1 (11.1) | 8 (19.0) |
| Anti–PD-1–based regimen as their most recent therapy | 4 (44.4) | 28 (66.7) |
| Refractory (PD during treatment or within 3 months after the last dose) | 9 (100) | 34 (81.0) |
| One line of prior PD-(L)1 blockade | 5 (55.6) | 20 (47.6) |
| PD-(L)1 monotherapy | 5 (55.6) | 14 (33.3) |
| PD-(L)1 combination therapy | 0 | 6 (14.3) |
| Two or more lines of prior PD-(L)1 blockade | 4 (44.4) | 21 (50.0) |
| Prior treatment line for Classical Hodgkin lymphoma | | |
| Median (Min-max) | 6.0 (3–9) | 3.5 (2–12) |

[a]Sex of participants was determined based on self-report.
BV, brentuximab vedotin.

In Part A (N = 25), the median age was 45 (range: 26–70) years old. The majority of the patients had Stage III/IV disease (96.0%) and an Eastern Cooperative Oncology Group performance status (ECOG PS) of 0 (84.0%). The median prior anti-tumor treatment line was 4, and 13 (52.0%) patients had received PD-(L)1 blockade. There were 9 (36.0%) patients with cHL with a median prior anti-tumor treatment line of 6, and all of them had received PD-(L)1 blockade (Table 1).

In Part B (N = 46), the median age was 36 (range: 19 to 68) years old. Most of the patients had Stage III/IV disease (82.6%) and an ECOG PS of 0 (56.5%). The median prior anti-tumor treatment line was 3, and 41 (89.1%) patients received prior PD-(L)1 blockade. There were 42 patients (91.3%) with cHL with a median prior anti-tumor treatment line of 3.5, and 41 of them had received PD-(L)1 blockade. Furthermore, among all the cHL patients, 34 patients were refractory to PD-(L)1 blockade (defined as disease progressing during treatment or within 3 months after the last dose), 28 patients received PD-1 blockade as their latest therapy, and 21 patients received at least 2 lines of PD-(L)1 blockades (Table 1). All the 7 patients who were not PD-(L)1 refractory to prior PD-(L)1 exposure had eventually relapsed.

## Safety

Incidence of adverse events, maximum tolerated dose (MTD), and recommended phase 2 dose (RP2D) of tifcemalimab as monotherapy and in combination with toripalimab were the primary endpoints.

The safety profile was evaluated among the 71 patients who had received at least one dose of study treatment. The median duration of treatment was 15.9 weeks (range: 3.0–56.9) in Part A and 33.6 weeks (range: 3.0–123.0) in Part B.

In Part A, no dose-limiting toxicities (DLTs) or deaths occurred. Maximum tolerated dose (MTD) was not observed. Treatment-related adverse events (TRAEs) occurred in 2 (2/3, 66.7%), 11 (11/12, 91.7%), 1 (1/3, 33.3%) and 6 (6/7, 85.7%) patients respectively in the 1 mg/kg, 3 mg/kg, 10 mg/kg and 200 mg cohorts; in total, 20 (20/25, 80.0%) patients experienced at least one TRAE as assessed by the investigator (Supplementary Table 1). The most common (incidence ≥ 20%) TRAE was anemia (20.0%) (Table 2). Grade 3/4 TRAEs occurred in 3 patients (12.0%), including gamma-glutamyl transferase increased, blood bilirubin increased, lymphocyte count decreased, and anemia (1 patient for each, 4.0%) (Supplementary Table 2). Serious adverse events (SAEs) occurred in 2 (8.0%) patients, and all were not treatment-related. Two (8.0%) patients experienced TRAEs leading to study treatment interruption, including 1 patient in the 1 mg/kg cohort with Grade 2 herpes zoster, and 1 patient in the 3 mg/kg cohort with both Grade 3 gamma-glutamyl transferase increased and Grade 3 blood bilirubin increased. No patients discontinued the study treatment due to AEs. Immune-related adverse events (irAEs) of Grade 3 or greater were not reported in Part A.

In Part B, 42 (42/46, 91.3%) patients experienced at least one TRAE (Supplementary Table 3). TRAE with an incidence ≥ 20% was anemia (23.9%) (Table 2). Grade 3/4 TRAEs occurred in 32.6% of the patients, with pyrexia and pneumonia occurring in 2 patients (4.3% for each) (Supplementary Table 4). Ten (21.7%) patients experienced study treatment-related SAEs, with no deaths reported due to SAEs. Adverse events leading to study drug discontinuation occurred in 6 (13.0%) patients, including Grade 4 anaphylactic shock, Grade 3 pneumonitis, Grade 4 sepsis, Grade 3 immune-mediated hepatitis, Grade 3 hepatic function abnormal and Grade 3 drug eruption (2.2% for each), and all these events were resolved after receiving concomitant treatments including corticosteroids (5 patients) and adrenocortical hormone (1 patient). TRAEs leading to treatment interruption occurred in 14 (30.4%) patients (Supplementary Table 5). Grade 3 or greater irAEs were reported in 5 patients (10.9%), including anaphylactic shock, pyrexia, and immune-mediated hepatitis, drug eruption, hepatic function abnormal, and lipase increase (1 patient, 2.2% for each).

**Table 2 | Treatment-Related Adverse Events (TRAEs) with an Incidence ≥ 10% (Full Analysis Set)**

| Preferred Term | Monotherapy (Part A) N = 25 n (%) | | | Combination therapy (Part B) N = 46 n (%) | | |
|---|---|---|---|---|---|---|
| | Grade 1-2 | Grade 3-4 | Total | Grade 1-2 | Grade 3-4 | Total |
| Anaemia | 4 (16.0) | 1 (4.0) | 5 (20.0) | 10 (21.7) | 1 (2.2) | 11 (23.9) |
| Pyrexia | 4 (16.0) | 0 | 4 (16.0) | 7 (15.2) | 2 (4.3) | 9 (19.6) |
| Sinus bradycardia | 0 | 0 | 0 | 8 (17.4) | 0 | 8 (17.4) |
| White blood cell count decreased | 4 (16.0) | 0 | 4 (16.0) | 7 (15.2) | 0 | 7 (15.2) |
| Sinus tachycardia | 2 (8.0) | 0 | 2 (8.0) | 7 (15.2) | 0 | 7 (15.2) |
| Hypothyroidism | 2 (8.0) | 0 | 2 (8.0) | 7 (15.2) | 0 | 7 (15.2) |
| Platelet count decreased | 2 (8.0) | 0 | 2 (8.0) | 7 (15.2) | 0 | 7 (15.2) |
| Upper respiratory tract infection | 1 (4.0) | 0 | 1 (4.0) | 6 (13.0) | 0 | 6 (13.0) |
| Pneumonia | 0 | 0 | 0 | 4 (8.7) | 2 (4.3) | 6 (13.0) |
| Rash | 0 | 0 | 0 | 6 (13.0) | 0 | 6 (13.0) |
| Hypotension | 0 | 0 | 0 | 6 (13.0) | 0 | 6 (13.0) |
| Hyperuricaemia | 0 | 0 | 0 | 5 (10.9) | 0 | 5 (10.9) |
| Activated partial thromboplastin time abnormal | 0 | 0 | 0 | 5 (10.9) | 0 | 5 (10.9) |
| Blood thyroid-stimulating hormone increased | 0 | 0 | 0 | 5 (10.9) | 0 | 5 (10.9) |
| Neutrophil count decreased | 0 | 0 | 0 | 5 (10.9) | 0 | 5 (10.9) |
| Hepatic function abnormal | 0 | 0 | 0 | 4 (8.7) | 1 (2.2) | 5 (10.9) |
| Asthenia | 3 (12.0) | 0 | 3 (12.0) | 4 (8.7) | 0 | 4 (8.7) |
| Aspartate aminotransferase increased | 3 (12.0) | 0 | 3 (12.0) | 3 (6.5) | 0 | 3 (6.5) |
| Lymphocyte count decreased | 2 (8.0) | 1 (4.0) | 3 (12.0) | 2 (4.3) | 0 | 2 (4.3) |

Treatment-related adverse events (TRAEs) were defined as TEAEs that were related or likely related to the study treatment as assessed by the investigator.

## Determination of RP2D

No DLTs were observed for tifcemalimab at the dose of 1 to 10 mg/kg, and the incidence of treatment-emergent adverse events (TEAEs) was similar across all the studied doses of tifcemalimab. The BTLA receptor occupancy remained consistently high (>80%) when tifcemalimab was administered at the dose of ≥1 mg/kg (Supplementary Table 6). In the 3 mg/kg cohort, one out of 12 patients achieved a partial response (PR), whereas no efficacy improvements were observed in patients receiving 10 mg/kg. These findings suggest that the 3 mg/kg dosage may be the optimal dose based on efficacy outcomes. Furthermore, the PK profile of tifcemalimab appeared to be similar between the 3 mg/kg and 200 mg cohorts. Considering the safety, pharmacokinetics, pharmacodynamics, and clinical activity findings, tifcemalimab 200 mg was selected as RP2D for the monotherapy. For combination therapy, tifcemalimab was well-tolerated when administered at both the doses of 100 mg and 200 mg with 240 mg toripalimab. Thus, tifcemalimab 200 mg was determined as the RP2D for both monotherapy and combination therapy.

## Pharmacokinetics and pharmacodynamics

The pharmacokinetics (PK) profiles of tifcemalimab under both single-dose and multi-dose regimens were characterized from 45 patients (Supplementary Table 7). The serum concentration of tifcemalimab after the first administration is shown in Supplementary Fig. 1. In Part A, drug exposure of tifcemalimab at the dose of 1, 3 or 10 mg/kg (maximum drug concentration [$C_{max}$] and area under curve [$AUC_{0-21d}$]) increased in a dose-proportional manner, and the half-life of the first cycle was $11.8 \pm 2.36$, $13.1 \pm 4.96$, and $17.6 \pm 1.05$ days, respectively; drug exposure of tifcemalimab after the first administration was similar between the 3 mg/kg and 200 mg cohorts, with $C_{max}$ of $60.6 \pm 19.2$ μg/mL and $67.6 \pm 10.8$ μg/mL, and $AUC_{0-21d}$ of $11923.4 \pm 3531.51$ h*μg/mL and $12542.2 \pm 2037.11$ h*μg/mL, respectively. Compared to the tifcemalimab monotherapy at the dose of 200 mg, the PK parameters of tifcemalimab (200 mg) administered in combination with toripalimab (240 mg) exhibited similar characteristics, which indicated toripalimab did not affect the metabolism of tifcemalimab. Meanwhile, tifcemalimab did not affect the toripalimab PK profile based on the trough concentration of

toripalimab. Full BTLA-receptor occupancy was maintained following the administration of tifcemalimab at the dose of 1–10 mg/kg or 200 mg (Supplementary Table 7). In addition, peripheral blood samples were collected from 31 patients in either the Part A or Part B cohorts before and 24 h after tifcemalimab administration, while no significant changes in effector cells related to efficacy were observed (Supplementary Table 8).

## Efficacy

In Part A (N = 25), the decrease of target lesions of any size from baseline was observed in 8 patients (8/24) (Supplementary Fig. 3A). One patient achieved partial response (PR), 6 had stable disease (SD), producing an objective response rate (ORR) of 4.0% (Supplementary Table 9). The median progression-free survival (PFS) was 1.9 months (95% confidence interval [CI]: 1.6, 2.1) (Supplementary Fig. 2A). Tumor response during treatment for individuals is provided in Supplementary Fig. 3B. Among the 7 cHL patients with prior anti-PD-(L)1 regimen treated at the dose of recommended phase 2 dose (RP2D) (200 mg) or the dose close to RP2D (3 mg/kg), 3 SD (disease control rate [DCR] 42.9%) were observed with an estimated median PFS of 1.9 months (95% CI: 1.6, 10.6) (Supplementary Table 9).

In Part B (N = 46), the decrease of target lesions size from baseline was observed in 37 patients (37/43) (Fig. 2A). One patient achieved complete response (CR), 16 achieved PR, and 20 had SD, resulting in an ORR of 37.0% and a DCR of 80.4%. The estimated median duration of response (DoR) was 12.5 months (95% CI: 6.1, 25.1) (Supplementary Table 9), and the median PFS was 13.1 months (95% CI: 6.4, 16.4), respectively (Supplementary Fig. 2B). Responses were still ongoing in 5 of the 17 responders. Eight patients in the 200 mg cohort and 5 in the 100 mg cohort continued therapy after progression, and all developed persistent PD except 1 in the 200 mg cohort who experienced PR after progression (Fig. 2B). In addition, in the RP2D cohort (tifcemalimab 200 mg and toripalimab 240 mg), 34 patients had received PD-(L)1 blockade. Among those patients, the decrease of target lesions size from baseline was observed in 28 (28/34, 82.4%) patients (Supplementary Fig. 3C). 12 PR (ORR 35.3%), 17 SD (DCR 85.3%) and 4 PD were observed, with an estimated median DoR of 6.3 months (95% CI: 2.4, 14.5) and a

A

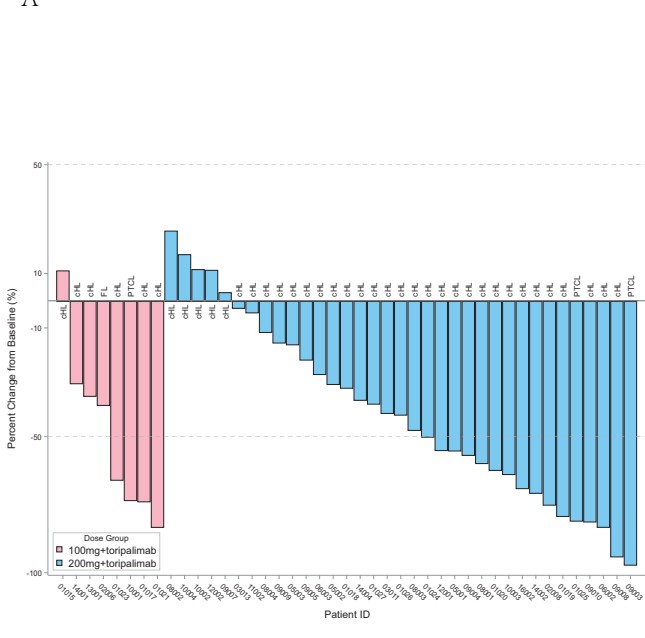

B

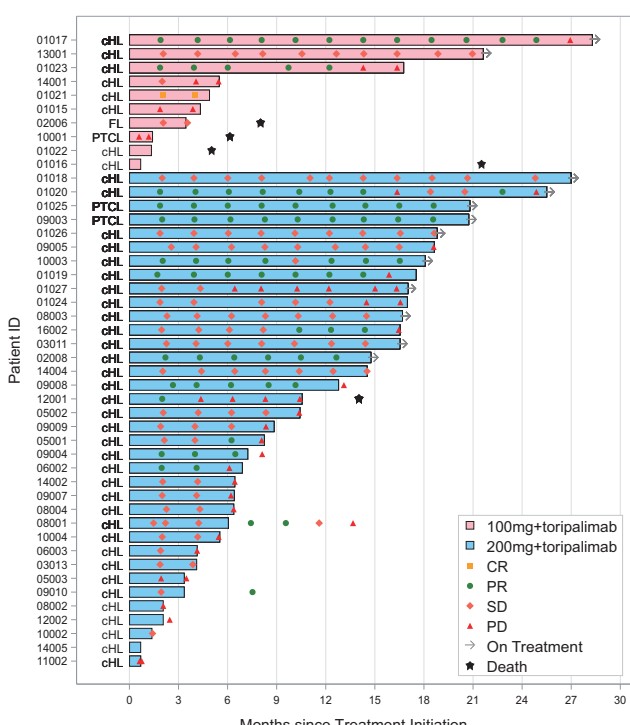

**Fig. 2 | Tumor response.** Figure (**A**) shows the best change of target from baseline after the administration of toripalimab 240 mg combined with tifcemalimab 100 mg and 200 mg (43 patients*). Figure (**B**) shows the tumor response after the administration of toripalimab 240 mg combined with tifcemalimab 100 mg and 200 mg (46 patients). * In Figure (**A**), 3 patients (01022, 01016, 14005) in Part B were not presented in the figure due to lack of post-baseline tumor evaluation data, while they were included in all other parameters as they had received at least one dose of the study drug and were included in the FAS (full analysis set) per the protocol. cHL = classical Hodgkin's lymphoma, DLBCL = diffuse large B-cell lymphoma, FL = follicular lymphoma, PTCL = peripheral T-cell lymphoma, PR = partial response, SD = stable disease, PD = progressive disease. Source data are provided as a Source Data file.

median PFS of 10.4 months (95% CI: 6.4, 15.9) (Supplementary Fig. 2C and Supplementary Table 9). Tumor response during treatment for individuals is provided in Supplementary Fig. 3D. In the RP2D cohort, there were 29 patients with cHL refractory to PD-(L)1 blockade, defined as the patients who had disease progression within 3 months after the last dose of PD-(L)1 blockade. The ORR for those patients was 31.0% (9 PR) with a median DoR of 8.4 months (95% CI: 2.4, 14.5), and the median PFS was 8.3 months (95% CI: 6.1, 15.9). In addition, there were 5 cHL patients in the RP2D cohort relapsed (defined as progression after 3 months from the last dose of PD-1) after PD-(L)1 blockade, and 3 of them responded to the study treatment, with an ORR of 60% (Supplementary Fig. 2D and Supplementary Table 9).

## Biomarker

Among the cHL patients who received tifcemalimab in combination with toripalimab, HVEM and PD-L1 biomarker data were available from 20 and 17 patients, respectively. High expression of both HVEM and PD-L1 were observed in patients with cHL. 90% (18/20) of the patients were HVEM positive, and 82% (14/17) were PD-L1 positive. In addition, the two patients who were HVEM-negative did not achieve a response. Thus, a trend that higher response rate in patients with positive HVEM expression was observed (Fig. 3). Supplementary Fig. 3 shows the representative staining of HVEM. The association of biomarker with clinical efficacy was not assessed for patients who received monotherapy and non-cHL who received combination therapy, due to limited response rate and small sample size, respectively, which would not lead to a clear conclusion. The patient-level data on HVEM and PD-L1 staining in patients with cHL and non-Hodgkin's lymphoma is provided in Supplemental Table 10.

## Discussion

No DLTs were observed in this Phase I study of tifcemalimab as monotherapy or combined with toripalimab in R/R lymphoma. Tifcemalimab was well tolerated at all the studied doses. The RP2D was selected as tifcemalimab 200 mg for both monotherapy and combination therapy based on the pharmacokinetics analyses, safety, and anti-tumor activity. The study also indicated that the combination of tifcemalimab and torpalimab yielded clinical benefits in cHL patients who had previously received PD-(L)1 blockade-containing regimens.

Tifcemalimab alone or in combination with toripalimab demonstrated favorable safety in patients with lymphoma. For the combination therapy, the majority of the TRAEs were Grade 1-2, and TRAE of Grade 3 or higher occurred in 32.6% of the patients. There were no life-threatening or fatal treatment-related events in this study. While in this study, the most common (≥20%) TRAE was anemia (23.9%). Previously, the reported most common (≥30%) TRAEs of other PD-1 products in cHL included upper respiratory tract infection, musculoskeletal pain, anemia, fatigue, cough, diarrhea, hyperglycemia, increased aspartate transaminase, etc, refs. 10–13. Except for anemia, the incidences of these TRAEs were less than 20% in the present study, suggesting that tifcemalimab plus toripalimab was safer compared with other PD-1 products in lymphoma. Generally, in cHL patients, the addition of tifcemalimab to toripalimab did not increase the severity and the incidence of adverse events compared to PD-1 monotherapies.

Meanwhile, many other novel treatments following PD-(L)1 blockade are still under investigation. A median PFS of 6.3 months was shown in a retrospective study involving cHL patients who progressed after PD-(L)1 inhibitors with available options including bendamustine, lenalidomide, evolimus, and gemcitabine or participation in clinical

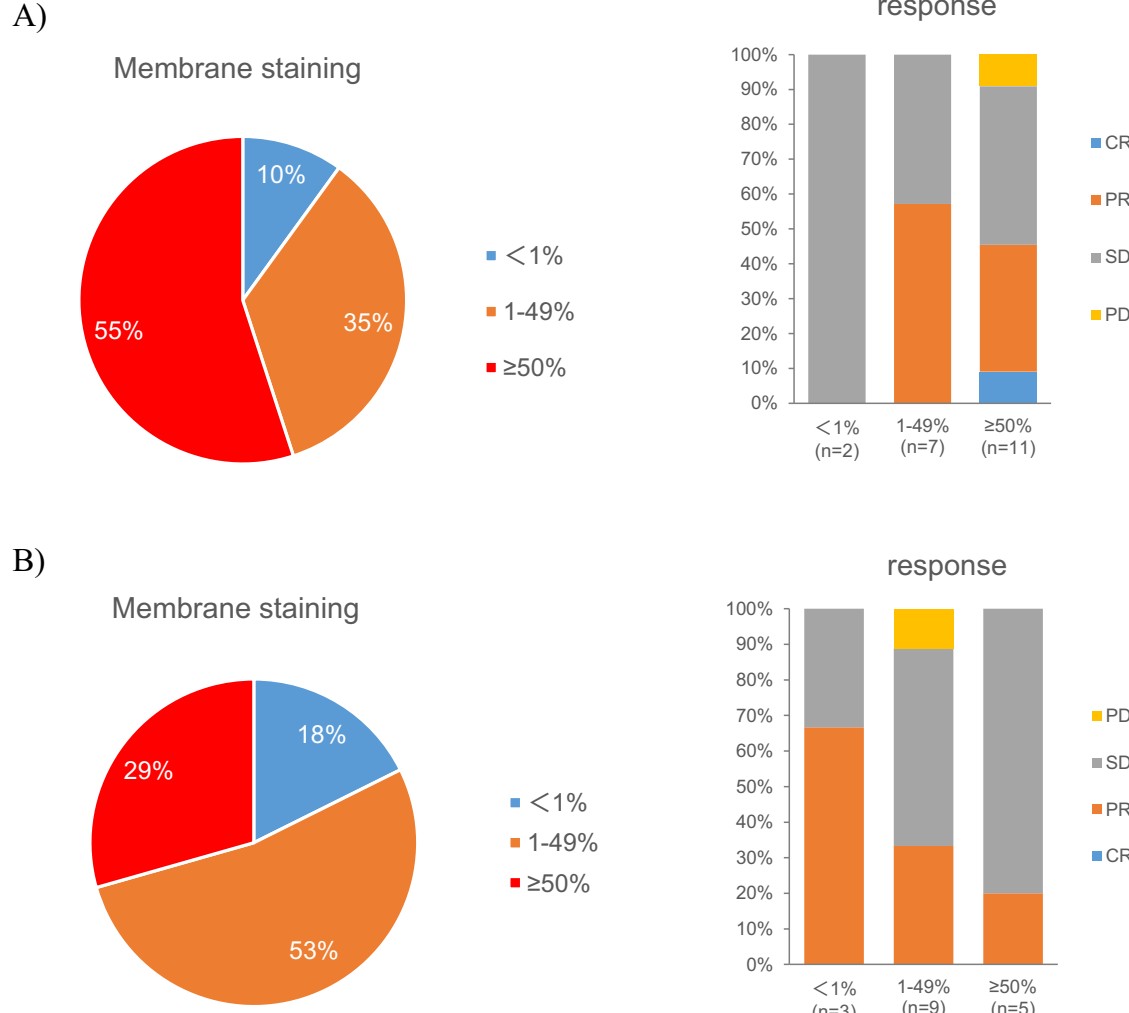

**Fig. 3 | Expression of Herpes Virus Entry Mediator and Programmed Death-Ligand.** Figure (**A**) shows the membrane staining score of HEVM and tumor response. Figure (**B**) shows the membrane staining score of PD-L1 and tumor response. CR = complete response, PR = partial response, SD = stable disease, PD = progressive disease. Source data are provided as a Source Data file. N means the number of patients.

studies[14]. Recently, several early phase studies of PD-(L)1 -based therapies have been explored in cHL patients who had previously received PD-(L) blockade showed promising outcomes, such as the study of favezelimab (anti-LAG-3) plus permbrolizumab[15], sabestomig (binding to PD-1 and T cell immunoglobulin and mucin-domain containing protein-3 [TIM-3])[16], CC-486 (hypomethylating agent) plus nivolumab[17], and decitabine plus camrelizumab[18]. The results of the first three studies showed an ORR of approximately 30% and a median PFS of about 11.0 months, which is consistent with the outcomes produced in our study. Whereas the treatment of cHL with decitabine plus camrelizumab showed the ORR of 52.0–68.0% and the median PFS of 20.0 -21.6 months (Study NCT02961101)[18], suggesting better outcomes compared to the present study (ORR 35.3%, PFS 10.4 months). The difference in the ORR and PFS is probably due to the patients enrolled in our study who were was more PD-(L)1 heavily pretreated, as we enrolled not only patients who received prior PD-(L)1 monotherapy, but also who received prior PD-(L)1-based combination therapies (14.3%), as well as patients with more than one line of PD-(L)1 blockade treatment (50.0%).

HVEM and PD-L1 positivity were not included in the eligibility criteria, but were collected for retrospective test. The results indicated that the expression of HVEM and PD-L1 is generally elevated in cHL,

with positive rates of 90% and 82%, respectively, which might explain the clinical activity in cHL. However, the relationship between HVEM, PD-L1 expression and clinical efficacy remains uncertain due to insufficient data.

The primary limitation of this study is the lack of controlled design; the second limitation is the limited sample size leading to the lack of robust evidence for a more definitive conclusion. A confirmatory Phase III study exploring tifcemalimab plus toripalimab combination in patients with relapsed or refractory cHL who have received a PD-(L)1 blockade is ongoing.

In conclusion, tifcemalimab as monotherapy or in combination with toripalimab demonstrated a favorable safety profile in lymphoma patients. Tifcemalimab in combination with toripalimab showed promising clinical efficacy among patients with classical Hodgkin's lymphoma who had previously received PD-(L)1 blockades.

## Methods

The study was conducted in compliance with the Declaration of Helsinki, China Good Clinical Practice (GCP), and applicable laws and regulations. The protocol and all amendments were approved by the ethics committee at ethics committee. A list of ethics committees is provided in the Supplementary Note 1. The protocol was preregistered

on July 17, 2020 (the preregistered protocol see Record History|ver. 1: 2020-07-17|NCT04477772|ClinicalTrials.gov).

## Patients

Eligible patients must have pathologically confirmed lymphoma (all subtypes of lymphoma, including peripheral T-cell lymphoma, Hodgkin lymphoma, follicular lymphoma or other subtypes) that was relapsed or refractory disease. Patients must have relapsed after autologous stem cell transplantation (ASCT) or not suitable for ASCT. Other key eligibility criteria were being 18–70 years old (inclusive), having an ECOG PS score of 0 to 1, hemoglobin at least 90 g/L, absolute neutrophil count at least $1.5 \times 10^9$/L, and platelet count at least $100 \times 10^9$/L. Key exclusion criteria included prior treatments with anti-BTLA or anti-HVEM antibodies, presence of unrelieved toxicities from prior anti-tumor therapies, presence of central nervous system disease that was symptomatic, untreated, or required continued treatment, and current autoimmune disease. The complete eligibility criteria are provided in the full Study Protocol (see Supplementary Note 2). Refer to Record History|NCT04477772|ClinicalTrials.gov for any protocol amendments made during the study.

## Procedures

The sample size for the dose-escalation stage was based on a 3 + 3 design. Since it was an early phase study, the sample size for indication expansion calculation was not based on a statistical hypothesis. Approximately 170 patients were planned to be enrolled into the study. The sample size would be adjusted based on the efficacy and safety data obtained during the study.

This phase I study consisted of Part A (tifcemalimab monotherapy) and Part B (combination therapy) and enrolled patients in China. In Part A, patients received tifcemalimab at the dose of 1, 3 or 10 mg/kg (every 3 weeks [Q3W]) intravenously for dose escalation (3 + 3 design) (3 to 6 patients were planned for each dose cohort), 3 mg/kg and a fixed dose of 200 mg (Q3W) for dose expansion (6 to 9 patients were planned for each dose cohort), and the RP2D for indication expansion (approximately 15 patients for each tumor subtypes, including peripheral T-cell lymphoma, Hodgkin lymphoma, follicular lymphoma or other subtypes). In Part B, patients received tifcemalimab 100 mg or 200 mg (Q3W) plus toripalimab (240 mg Q3W intravenously) for dose-escalation (3–6 patients were planned for each dose cohort), tifcemalimab at recommended dose plus toripalimab for dose expansion (6–9 patients were planned for each dose), and tifcemalimab at RP2D plus toripalimab for indication expansion (approximately 20 patients for each tumor subtype including peripheral T-cell lymphoma, Hodgkin lymphoma, follicular lymphoma or other subtypes). The study treatments could be continued until intolerable toxicity, disease progression, or death, withdrawal of consent by patients, or up to 2 years of treatment with tifcemalimab or toripalimab. Patients who experienced the progression as defined by Lugano 2014 Criteria might continue treatment but would undergo repeat imaging within 4–8 weeks (± 7 days) for confirmation of progression if they met the following conditions: (1) No clinically significant symptoms or signs of progressive disease; (2) No decrease in ECOG performance status; (3) No symptomatic rapid progression that required urgent medical intervention (such as symptomatic pleural effusion or spinal cord compression, etc.). Please refer to Supplementary Note 2 for details of the study design in the study protocol.

The primary endpoints included the incidence of adverse events and MTD/RP2D of tifcemalimab as monotherapy and in combination with toripalimab. Secondary endpoints included the clinical efficacy (include ORR, DoR, DCR, PFS, and OS), pharmacokinetics, pharmacodynamic (BTLA receptor occupancy on $CD3^+$, $CD4^+CD45RA^+$, $CD8^+CD45RA^+$, $CD3^-CD20^+$, and $CD3^-CD56^+$ cells) of tifcemalimab as monotherapy and in combination with toripalimab, and effects of tifcemalimab on peripheral blood cytokines (IFN-γ, IL-8, IL-4, IL-6, IL-10, and TNF-α).

DLTs were to be collected for 21 days after the first dose of the study treatment during dose escalation. DLT was defined as Grade ≥ 3 or 4 hematologic toxicities, Grade ≥ 3 non-hematological toxicities, death that could not be clearly attributed to progressive disease or other extraneous factors, toxicities leading to premature termination of the study, or adverse events (AEs) leading to treatment interruption or discontinuation. All AEs including SAEs, were captured through 90 days after the last dose of study treatments or until initiation of a new anti-tumor therapy, whichever occurred first. AEs were graded according to the National Cancer Institute-Common Terminology Criteria for Adverse Events (NCI-CTCAE) version 5.0.

Tumor status was evaluated by the investigators per 2014 Lugano criteria every 9 weeks (± 7 days) after the first dose of study treatment until disease progression, initiation of new anti-tumor therapy, withdrawal of consent, lost to follow-up, or death, whichever occurred first. Tumor imaging examination methods included enhanced computed tomography (CT) or magnetic resonance imaging (MRI). Baseline FDG-PET examination was required within 28 days before enrollment. During the treatment, an FDG-PET examination was to be performed within 4 weeks when CR was assessed by imaging such as CT or MRI.

Biological sample collection for PK and BTLA receptor occupancy was detailed in the study protocol.

Archival or fresh tumor biopsy samples were obtained from patients prior to treatment. HVEM expression was determined using a validated immunohistochemistry (IHC) assay at Labcorp with anti-HVEM mouse monoclonal antibody (clone 122, BioLegend, Inc.). A lymphoma score (the positively staining cells over all the nucleated cells within the tumor area) reported by the pathologist was used to evaluate HVEM expression. PD-L1 expression was determined by IHC assay staining with JS311 antibody validated in a central laboratory (MEDx). The distribution of both HVEM and PD-L1 expression was classified into: <1% (negative), 1–49% (low), and ≥ 50% (high).

## Study oversight

The study was sponsored by Shanghai Junshi Biosciences Co., Ltd. Tifcemalimab and toripalimab were manufactured and supplied by Shanghai Junshi Biosciences. During the study, due to the modest efficacy outcomes of tifcemalimab monotherapy in lymphoma (Part A), the study protocol was amended to investigate the combination therapy of tifcemalimab plus toripalimab in lymphoma (Part B) after discussion with the investigators, and the sample size was modified accordingly. Written informed consent was obtained from all patients. The first patient was enrolled on July 17, 2020, and the last patient was enrolled on January 28, 2023.

The study was overseen by the Sponsor and the ethics committee at each study site. Data were collected and monitored by the investigators and study personnel using an Electronic Data Capture System (EDC).

## Statistical methods

Patients who received at least one dose of study treatment were evaluable for safety analysis and efficacy analysis. Terms of adverse events were coded using MedDRA, version 23.0. The incidence and severity of TEAEs and TRAEs were summarized with descriptive statistics for Part A and Part B, respectively. SAS 9.4 was employed to calculate descriptive statistics. Efficacy endpoints, including PFS, ORR, DCR, and DoR, were assessed for Part A and Part B, and for patients who had R/R cHL previously treated with PD-(L)1 blockade, respectively. The Kaplan-Meier method was used to estimate the median and quantiles of PFS. Brookmeyer Crowley method, with log-log transformation applied to the survival function, was used to construct the associated 95% CI. Refer to the statistical analysis plan for detailed information on safety and efficacy analysis.

Patients' intensive serum concentration data of tifcemalimab were used to calculate the relevant PK parameters using WinNonlin

Version 8.3 (Certara) software with a non-compartmental Analysis model. The mean concentration-time curve of cycle 1 was plotted.

## Reporting summary
Further information on research design is available in the Nature Portfolio Reporting Summary linked to this article.

## Data availability
The study protocol was provided as a Supplementary Note 2. All requests for individual participant data will be reviewed by the leading clinical site, Peking University Cancer Hospital & Institute, and the study sponsor, Shanghai Junshi Biosciences, to verify whether the request is subject to any patient privacy, intellectual property, or confidentiality obligations. Requests for access to the patient-level data from this study for research purposes can be submitted via email to zhu-jun2017@outlook.com with detailed proposals for approval. A signed data access agreement with the sponsor is required before accessing shared data. Access is provided after a proposal has been approved by an independent review committee identified for this purpose and after receipt of a signed data sharing agreement. Access to all individual participant data collected during the trial will be provided after anonymization. Data and documents will be provided in a secure data-sharing environment. No expiration date of data requests is currently set once data are made available. Source data are provided with this paper. All remaining data can be found in the Article, Supplementary, and Source Data files. Source data are provided in this paper.

## Code availability
No custom code was used for statistical analysis in this study.

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

## Acknowledgements
The study was sponsored by Shanghai Junshi Biosciences Co., Ltd, and supported by Shanghai Science and Technology Innovation Action Plan Special Project for Biomedical Technology Support (Grant No.: 22S11903000). Junshi is responsible for study oversight, data collection, data interpretation and writing, and reviewing of the manuscript, and submission of the manuscript for publication. We thank all patients who participated in this study and their families, as well as all investigators involved in this study for their contributions. We particularly recognize Peng Xue, the clinical project manager, for his leadership in driving the research forward, ensuring patient recruitment, and upholding the highest standards of study quality. This manuscript was drafted by the medical writer (Lan Fang, Shanghai Junshi Biosciences Co., Ltd.) with the consent of all authors.

## Author contributions
**Conceptualization and design:** J.Z., Y.S., J.M, Y.W., W.W. and J.X. **Methodology:** J.Z., Y.S., Y.W., W.W. and J.X. **Provision of study materials or patients:** J.Z., Y.S., J.M., Y.X., H.Z., Y.S., Z.P., F.L., Y.H., W.Z. and W.H. **Data curation:** J.Z., Y.S., Y.W., W.W., R.D., Q.M. and J.X. **Formal Analysis:** J.Z., Y.S. and J.X. **Writing – review & editing:** All authors. **Final approval of manuscript:** All authors.

## Competing interests
Yifan Wang, Weiwei Wang, Jing Xu, Rong Deng, Qin Meng are employees of Shanghai Junshi Biosciences Co., Ltd. Other authors declare no potential competing interests for the submitted work.
