## [Transparent Peer Review file · Nature Communications]

Tifcemalimab as Monotherapy and in Combination with Toripalimab in Patients with Relapsed/Refractory Lymphoma: Phase I Trial

Corresponding Author: Dr Jun Zhu

Version 1:

Reviewer comments:

Reviewer #1

(Remarks to the Author)

The trial reports on tifcemalib (anti-BTLA antibody) and the combination of anti-BTLA4 + anti-PD1 antibodies in lymphoma, which is novel. The combination appears well-tolerated and Part A (N=25), treats a heterogenous group of NHL (8 FL, 9cHL, 2PTCL, 6 DLBCL) with tifcemalib monotherapy (anti-BTLA4 IgG4). Part B (N=46), describes dual therapy of tifcemalib and toripalimab for cHL. Notably, nearly all had received PD2/PD-L1 blockade (41/46), 81% of patients were refractory. Primary endpoint was safety and RP2D of tifce/tor.

The results are somewhat disappointing (only one-third) of patients responded compared to other reports of PD1+other agents such as vorinostat (Mei Blood 2023; ORR 65% in a similar population), although they did observe ~80% disease control.

- 1) Can the authors provide any further mechanistic insight aside from HVEM expression (very small numbers) of variables associated with response? Did they see an increase in tumor-specific cytotoxic T cells or effector cytokines, for instance? These types of observations would increase the impact of this manuscript.
- 2) Can the authors also report the ORR for those patients who did not have prior PD1/PDL1 therapy? This would place findings in better context. It is also difficult to know how to interpret the population of patients "pretreated with PD1/PDL1i." It would be very helpful to view the responses to PD1 treatment for patients refractory to prior PD1/PDL1 vs those patients relapsed afterward. It is well-documented that patients who previously had a CR to PD1i, can be retreated with response in a majority of cases.
- 3) More detail regarding AEs leading to drug discontinuation would be appreciated in the text. In addition, in Figure 1A: please clarify why treatment was discontinuous in dose expansion. Also, Figure 1B: reports intolerable toxicity? Please clarify the nature of this as well as other reasons for discontinuation.
- 4) I find the figures in the supplemental manuscript much more informative in interpreting the authors' results. Figures S2, S3, S4 were very valuable and I would encourage replacing most KMs and swimmer plots or tables in the main manuscript with these.
- 5) With regard to the PFS curves, it is probably not scientifically meaningful to provide PFS data for Cohort A given the myriad of dosing and histologies. Would only report PFS, would consider removing the waterfall plots and moving S2, S3, S4 into this area.
- 6) Please comment on the rationale for tumor imaging every 9 weeks and how this trial managed pseudoprogression in the methodology.
- 7) Please provide more details regarding how the RP2D was selected when toxicity seemed manageable across all dose levels and PKs and occupancy seemed similar across many doses?

8) Small point: please confirm "pre-treated with PD-1/PD-L1" means that these patients had previously received PD-1/PD-L1. If yes, please change language throughout the manuscript to patients who had received prior PD-1/PD-L1.

Reviewer #2

(Remarks to the Author)

This phase I study evaluated the safety and efficacy of Tifcemalimab as Monotherapy or in Combination with Toripalimab in Patients with Relapsed/Refractory Lymphoma. I have some comments:

1. This study modified the protocol on 15-Feb-2022 (according to the protocol version history) and on 13-Apr-2022 updated the registration at www.clinicaltrials.gov. The modification added Part B of this study. For the phase I trial, it's acceptable to modify the protocol according to the results of the study, but I recommend that the authors clearly state the protocol amendment in the manuscript and give some details on the reason for and process of the decision. In the protocol, the details of modification refer to " See Annex of Amendment Description for details", but I didn't find the Annex of Amendment.
2. An interim analysis results of this study were published in the 2022 ASCO Annual Meeting, available at " The Journal of Clinical Oncology, Volume 40, Number 16_suppl, https://doi.org/10.1200/JCO.2022.40.16_suppl.7578", in this published abstract, it stated that "The most common TEAEs were anemia (29.0%) and fever (22.6%).", but in the present manuscript, the TEAE of fever be removed, I would like to know the reason.
3. For Figure 1, reporting the number and reason for discontinuing the study treatment for each dose arm will be more clear and informative.
4. "The primary endpoints were to evaluate the safety and to determine the RP2D ", from my point of view "to evaluate the safety and to determine the RP2D" is the objectives, not the endpoints, need to improve the statement.
5. For Part B indication expansion, only cHL patients be included and the indication expansion for other subtypes was not conducted due to poor tumor response. From my point of view, this information is important and better be added to the Abstract.

Reviewer #3

(Remarks to the Author)

Song, Ma, et al present results of a phase 1 clinical trial of tifcemalimab with or without toripalimab in patients with relapsed/refractory lymphoma. Patients received tifcemalimab alone (Part A) or in combination with a PD1 inhibitor (Part B). They saw modest results with single agent tifcemalimab but in combination with PD1 inhibitor, saw promising early results in patients with Hodgkin lymphoma even though the majority of those patients were previously treated (and many refractory) to PD1 inhibitors. This suggests that this may potentially be a mechanism to overcome resistance in patients with PD1-refractory Hodgkin lymphoma. This is an interesting study in a high risk population and warrants further investigation in a larger cohort.

I have several comments:

Abstract: Would include a little more rationale in abstract. Would also include that toriplimab is a PD1 inhibitor as some readers may not be familiar with it.

Background: In some areas, PD1 antibodies are being used earlier in treatment of Hodgkin lymphoma (first line, salvage prior to autoSCT). Consider clarifying this in introduction.

Methods (Patients): Consider specifying that all subtypes of lymphoma included as well as subtypes of interest for study. Line 109 - "presence of central nervous system disease"

Results (Patients): Consider including information on patients who were not PD1 refractory with prior PD1 exposure (had they eventually relapsed, stopped therapy due to toxicity, or just stopped therapy)?

Discussion: What type of agent is CC-486 (line 316)?

Did some patients continue therapy past progression (as seen in Figure F)?

Reviewer #4

(Remarks to the Author)

Song et al report on a phase 1 trial using Toripalimab, an anti-BTLA antibody in 71 patients. The trial is broken up into two groups with the first one evaluating safety with monotherapy and a second cohort with Toripalimab and an anti-PD1 antibody, tifcemalimab.

There are not a lot of Anti-BTLA drugs so this study is important as this new drug is in testing and per article, additional studies. Based on the data listed here, the combination did seem to help, and this included some that failed checkpoint inhibitors in the past.

The trial design/methodology makes sense and was appropriate.

I don't see any flaws in data analysis.

The description of the trial was lacking a few details though.

1) It states no DLTs were found but the DLT criteria was not described. There were grade 3/4 hematologic AE's. The results

listed appropriate AE's/TRAE's/IRAE's and with the other PD1 inhibitors, this seemed to be about the same.

2) The trials seemed to heavily favor Hodgkin lymphoma's efficacy and highlights limited/no activity in NHL. The biomarkers described in the procedures section seems like they would test all disease types but they do not report per disease type and only report on CHL HVEM and PD-L1. This would also help in future studies or reproducibility.

3) Description of other trial sites, number of other sites.

4) Statistics. "The sample size for indication expansion was to be adjusted based on the efficacy and safety data obtained during the study," seems vague. Was this a 3x3 or boin design?

Version 2:

Reviewer comments:

Reviewer #1

(Remarks to the Author)

The authors have improved the manuscript with these revisions. The following comments remain:

Comment #1 follow-up: Thank you for providing this table. Please provide these data for T effector cells and cytokines in the supplement and make a note in the manuscript that this was assessed.

Comment #2 follow-up: Thank you for providing this information. Please add the ORR for relapsed cHL patients to the paragraph that details refractory patient responses. Please also add this information to table S7.

Comment #3 follow-up: Thank you for the additional drug hold and toxicity data. Please indicate what "all these events were resolved with concomitant treatments" means – ie clarify if this was managed with corticosteroids or something else. Please also add the reason for drug discontinuation to Figure 1 when this was due to an adverse event. Please report TRAEs (Table 2) as grade 1-2, grade 3-4, and total. This will help the reader understand the severity of these AEs. Please report Table S2 with the specific events and grades 1-2 vs 3-4 and total that occurred for each AE type. Please add to table S4 with the specific events graded 1-2 vs 3-4, and total events that occurred for event – it is important for understanding toxicity to also have the lower grade AE details. Please also indicate what events led to dose interruptions in the table.

Clarifying comments:

A) Figure 2B legend states: "One patient in Part A and 3 patients in Part B without baseline tumor evaluation data were not included in the figure." Can the authors clarify if these patients were included in any of the the efficacy assessments mentioned in the manuscript? Please also amend the number of efficacy evaluable patients to the manuscript text, as the text lists N=46.

B) Please add to the manuscript the number of patients who continued past progression as well as their clinical outcomes.

C) Please report the patient level data for all HVEM and PD-L1 staining, including both cHL and NHL in the supplement. I agree with reviewer #4 that histology also should be added even if small sample size.

D) With regard to reviewer #4, comment 3, the authors need to add the sample size information to the methods and also clarify what motivated the total number of patients enrolled.

Reviewer #2

(Remarks to the Author)

I have no further comments

Reviewer #3

(Remarks to the Author)

The authors have addressed my comments.

Reviewer #4

(Remarks to the Author)

My comments have been addressed. No additional comments.

Version 3:

Reviewer comments:

Reviewer #3

(Remarks to the Author)

The authors have adequately addressed all of the reviewer comments.

I have a few minor additional comments.

For Figure 1, the numbers in a few of the boxes don't quite add up. For Part A, for 3 mg/kg dose, the box states that treatment

discontinued in 8 patients but the reasons add up to 9. For the 200 mg dose, the box states treatment discontinued in 6 patients but the reasons add up to 7 (total number of patients treated). In part B, for the box Tifcemalimab 200 mg + toripalimab 240 mg, the box states that 5 patients discontinued treatment but there are only 2 patient reasons given. Small typo in Figure 1 legend, should be "as of study completion." Would also change "hepatic function abnormal" to "abnormal hepatic function."

Would clarify what is meant by lack of post tumor baseline evaluation data since these patients were otherwise evaluable for all other parameters (this refers to the 1 patient in Part A and 3 patients in part B who didn't have post baseline evaluation). Would state more explicitly why evaluable for response but not target lesion evaluation.

Figure 2E was a little confusing to me. If a patient is still on treatment, why were there responses shown past the arrow? I also couldn't find the patient who had partial response after progressive disease in part B.

A point-by-point response to the reviewers' comments:

Reviewer #1:

The trial reports on tificemalib (anti-BTLA antibody) and the combination of anti-BTLA4 + anti-PD1 antibodies in lymphoma, which is novel. The combination appears well-tolerated and Part A (N=25), treats a heterogenous group of NHL (8 FL, 9cHL, 2PTCL, 6 DLBCL) with tificemalib monotherapy (anti-BTLA4 IgG4). Part B (N=46) describes dual therapy of tificemalib and toripalimab for cHL. Notably, nearly all had received PD2/PD-L1 blockade (41/46), 81% of patients were refractory. Primary endpoint was safety and RP2D of tifice/tor.

The results are somewhat disappointing (only one-third) of patients responded compared to other reports of PD1+other agents such as vorinostat (Mei Blood 2023; ORR 65% in a similar population), although they did observe ~80% disease control.

Response: Thank you for the comments. We acknowledge that the ORR of our study is numerically lower than that of pembrolizumab plus vorinostat study. However, given the small sample size and different baseline characteristics of patients between the two studies, such comparison might be inappropriate. The proportion of patients with cHL who had received prior PD-1 blockade at baseline is higher in our study than that in the vorinostat study (97.6% vs. 78%), and half of the patients in our study had received ≥ 2 L of PD-1-based regimen, suggesting our patients might receive more adequate previous PD-1 treatment. Additionally, positron emission tomography (PET)-computed tomography (CT) scans were used to assess responses in the vorinostat study, while in our study, we mainly used CT for responses evaluation, which was likely less sensitive than PET-CT. All factors mentioned above might potentially affect the outcomes of Tificemalimab treatment.

1) Can the authors provide any further mechanistic insight aside from HVEM expression (very small numbers) of variables associated with response? Did they see an increase in tumor-specific cytotoxic T cells or effector cytokines, for instance? These types of observations would increase the impact of this manuscript.

Response: Thank you for the suggestions. No significant changes in effector cells related to efficacy were observed among the 45 participants whose peripheral blood cell subsets were analyzed. Peripheral blood samples were collected from 31 patients in either the Part A or Part B groups of the study before and 24h after administration. The table below presents the ratio of cytokines concentration 24 hours post-administration compared to pre-dose levels for each dose group. The results indicated varying degrees of cytokine elevation appeared 24 hours after initial administration in Part A and Part B groups, with a slightly higher ratio observed in the Part B group compared to the Part A group, this observation should be further confirmed due to limited sample size.

Table 1: The ratio of cytokines concentration 24 hours post-administration compared to pre-dose levels

Groups	Doses	N	IFN- γ	IL-6	IL-8	IL-10	TNF- α
Part A	1mg/kg	3	1.59 \pm 0.37	2.19 \pm 0.85	1.51 \pm 0.39	1.22 \pm 0.2	1.12 \pm 0.22
	3mg/kg	12	1.82 \pm 1.46	2.43 \pm 3.99	1.36 \pm 0.51	1.13 \pm 0.16	1.08 \pm 0.16
	10mg/kg	3	1.17 \pm 0.12	5.56 \pm 5.64	2.25 \pm 1.22	1.13 \pm 0.21	1.13 \pm 0.28

	200mg	6	1.43±0.41	1.11±0.28	1.12±0.48	1.19±0.25	1.09±0.06
Part B	100mg	4	2.17±1.51	2.71±2.08	3.04±0.87	3.35±3.63	1.36±0.1
	200mg	3	6.21±5.15	5.29±3.56	1.35±0.51	15.96±16.67	1.74±0.37

2) Can the authors also report the ORR for those patients who did not have prior PD1/PDL1 therapy? This would place findings in better context. It is also difficult to know how to interpret the population of patients “pretreated with PD1/PDL1i.” It would be very helpful to view the responses to PD1 treatment for patients refractory to prior PD1/PDL1 vs those patients relapsed afterward. It is well-documented that patients who previously had a CR to PD1i, can be retreated with response in a majority of cases.

Response: Thank you for the suggestions. There was only 1 patient with Classical Hodgkin lymphoma in the 100mg dose group who did not have prior PD1/PDL1 therapy in our study. This pt had achieved CR but withdrew from the trial after twice efficacy assessments. In our study, PD-1 refractory was defined as disease progress within 3m from the last dose of PD-1. Progression after 3 m from the last dose of PD-1 was defined as relapses. The definitions referred to other cHL trials (such as NCT05508867 & NCT05162976). According to the definitions, among cHL patients who had received PD-(L)1 blockade in the part B RP2D cohort, 5 pts were classified to PD-1 relapsed, 3 of them responded, 29 pts were classified to PD -1 refractory, 9/29 pts responded. It seemed that patients relapsed after PD1/PDL1 therapy responded better than PD -1 refractory patients, and the finding was consistent with that reported in the similar study (Mei Blood 2023).

3) More detail regarding AEs leading to drug discontinuation would be appreciated in the text. In addition, in Figure 1A: please clarify why treatment was discontinuous in dose expansion. Also, Figure 1B: reports intolerable toxicity? Please clarify the nature of this as well as other reasons for discontinuation.

Response: Thank you for the suggestions. We have added more detail regarding AEs leading to drug discontinuation in the text (Page 8, Line 166-170), and revised Fig 1 (Page 25, 26).

4) I find the figures in the supplemental manuscript much more informative in interpreting the authors’ results. Figures S2, S3, S4 were very valuable, and I would encourage replacing most KMs and swimmer plots or tables in the main manuscript with these.

Response: Thank you for the suggestions. We have replaced Figure 2A, 2B, 2D, 2E and 2F with Figures S2, S3, S4 in the main manuscript (Page 27-31).

5) With regard to the PFS curves, it is probably not scientifically meaningful to provide PFS data for Cohort A given the myriad of dosing and histologies. Would only report PFS, would consider removing the waterfall plots and moving S2, S3, S4 into this area.

Response: Thank you for the suggestions. We have removed the PFS curve for Cohort A (Fig 2A) as well as the waterfall plot (Fig 2C), and moved S2, S3, S4 into this area (Page 27-31).

6) Please comment on the rationale for tumor imaging every 9 weeks and how this trial managed pseudoprogression in the methodology.

Response: Thank you for the comments. The rationale for tumor imaging every 9 weeks referred to other similar studies. We managed pseudo progression as described below:

Subjects who experience the progression as defined by Lugano 2014 Criteria may continue treatment but should undergo repeat imaging within 4-8 weeks (\pm 7 days) for confirmation of progression if they meet the following conditions:

- a. No clinically significant symptoms or signs of progressive disease;
- b. No decrease in Eastern Cooperative Oncology Group (ECOG) performance status;
- c. No symptomatic rapid progression that requires urgent medical intervention (such as symptomatic pleural effusion or spinal cord compression, etc.).

For subjects who meet the above criteria, the informed consent must be obtained again before continuing the treatment at the time of initial radiographic progressive disease. The informed consent form should fully inform all available standard therapies and the risks and benefits of continuing the treatment with the study drug. Sufficient communication and discussion should be made between the investigators and the sponsor, as well as between the investigators and the subjects. The study can be conducted only after they consider that the benefits of continued treatment outweigh the risks.

And we have added the relative information to the Methods section (Page 15, Line 319-324).

Reference:

Philippe Armand, et al. Programmed Death-1 Blockade With Pembrolizumab in Patients With Classical Hodgkin Lymphoma After Brentuximab Vedotin Failure. *J Clin Oncol*. 2016 Nov 1;34(31):3733-3739.

Dai Maruyama, et al. Multicenter phase II study of nivolumab in Japanese patients with relapsed or refractory classical Hodgkin lymphoma. *Cancer Sci*. 2017 May;108(5):1007-1012.

7) Please provide more details regarding how the RP2D was selected when toxicity seemed manageable across all dose levels and PKs and occupancy seemed similar across many doses?

Response: Thank you for the comments. It is true that PKs and occupancy data were similar across many doses in our study. In monotherapy group, a partial response was observed in the 3mg/kg group. Therefore, 0.3mg/kg was considered to be the pharmacological dose achieved, 3 mg/kg was considered to be the efficacious dose achieved, and an increase in dose (10mg/kg) did not result in a greater benefit for the patients. Furthermore, pharmacokinetics profiles were similar between the 3mg/kg and 200mg dose groups. So, 200mg dose level was considered as the appropriate RP2D after comprehensive evaluation.

8) Small point: please confirm “pre-treated with PD-1/PD-L1” means that these patients had previously received PD-1/PD-L1. If yes, please change language throughout the manuscript to patients who had received prior PD-1/PD-L1.

Response: Thank you for the suggestion. We confirm that “pre-treated with PD-1/PD-L1” means that these patients had previously received PD-1/PD-L1, and we have changed “pre-treated with PD-1/PD-L1” to “patients who had received prior PD-1/PD-L1” throughout the manuscript (Page 10, Line 201-202; Page 12, Line 244; Page 12, Line 257-258; Page 13, Line 284-285).

Reviewer #2

This phase I study evaluated the safety and efficacy of Tifcemalimab as Monotherapy or in Combination with Toripalimab in Patients with Relapsed/Refractory Lymphoma. I have some comments:

1. This study modified the protocol on 15-Feb-2022 (according to the protocol version history) and on 13-Apr-2022 updated the registration at www.clinicaltrials.gov. The modification added Part B of this study. For the phase I trial, it's acceptable to modify the protocol according to the results of the study, but I recommend that the authors clearly state the protocol amendment in the manuscript and give some details on the reason for and process of the decision. In the protocol, the details of modification refer to " See Annex of Amendment Description for details", but I didn't find the Annex of Amendment.

Response: Thank you for the suggestion. We have clearly stated the protocol amendment and gave some details on the reason for and process of the decision in the revision (Page 17, Line 360-364). Additionally, the annex of amendment for the protocol was uploaded with resubmission.

2. An interim analysis results of this study were published in the 2022 ASCO Annual Meeting, available at " The Journal of Clinical Oncology, Volume 40, Number 16_suppl, https://doi.org/10.1200/JCO.2022.40.16_suppl.7578", in this published abstract, it stated that "The most common TEAEs were anemia (29.0%) and fever (22.6%). ", but in the present manuscript, the TEAE of fever be removed, I would like to know the reason.

Response: Sorry for the confusion. We used “pyrexia” instead of “fever” in this manuscript, as listed in Table 2 (Page 24), pyrexia was the second common TRAE occurred in 19.6% of the combination therapy group.

3. For Figure 1, reporting the number and reason for discontinuing the study treatment for each dose arm will be more clear and informative.

Response: Thank you for the suggestion. We have revised Fig 1 to report the number and reason for discontinuing the study treatment for each dose arm (Page 25, 26).

4. "The primary endpoints were to evaluate the safety and to determine the RP2D ", from my point of view "to evaluate the safety and to determine the RP2D" is the objectives, not the endpoints, need to improve the statement.

Response: Thank you for the suggestion. We have revised "The primary endpoints were to evaluate the safety and to determine the RP2D" to “The primary endpoints included the incidence of adverse events and MTD/RP2D” (Page 15, Line 326-327).

5. For Part B indication expansion, only cHL patients be included and the indication expansion for other subtypes was not conducted due to poor tumor response. From my point of view, this information is important and better be added to the Abstract.

Response: Thank you for the suggestion. We have added this information to the Abstract (Page 4, Line 63-65).

Reviewer #3

Song, Ma, et al present results of a phase I clinical trial of tifcemalimab with or without toripalimab in patients with relapsed/refractory lymphoma. Patients

received tivecimalimab alone (Part A) or in combination with a PD1 inhibitor (Part B). They saw modest results with single agent tivecimalimab but in combination with PD1 inhibitor, saw promising early results in patients with Hodgkin lymphoma even though the majority of those patients were previously treated (and many refractory) to PD1 inhibitors. This suggests that this may potentially be a mechanism to overcome resistance in patients with PD1-refractory Hodgkin lymphoma. This is an interesting study in a high risk population and warrants further investigation in a larger cohort.

Response: Thank you for your recognition of our work and positive comments.

I have several comments:

Abstract: Would include a little more rationale in abstract. Would also include that toriaplimab is a PD1 inhibitor as some readers may not be familiar with it.

Response: Thank you for the suggestion. We have included a little more rationale, as well as that toriaplimab is a PD1 inhibitor in the Abstract (Page 4, Line 57-58).

Background: In some areas, PD1 antibodies are being used earlier in treatment of Hodgkin lymphoma (first line, salvage prior to autoSCT). Consider clarifying this in introduction.

Response: Thank you for the suggestion. We have clarified this in the Introduction section (Page 5, Line 85).

Methods (Patients): Consider specifying that all subtypes of lymphoma included as well as subtypes of interest for study. Line 109 - "presence of central nervous system disease"

Response: Thank you for the suggestion. We have specified that all subtypes of lymphoma included as well as subtypes of interest for study in Methods section (Page 14, Line 289-291). The lapsus calami "presence of central nervous system" has been corrected to "presence of central nervous system disease" (Page 14, Line 297).

Results (Patients): Consider including information on patients who were not PD1 refractory with prior PD1 exposure (had they eventually relapsed, stopped therapy due to toxicity, or just stopped therapy)?

Response: Thank you for the suggestion. A total of 7 patients in part B who were not PD-(L)1 refractory with prior PD-(L)1 exposure. 5 patients were in the 200mg cohort: they all eventually relapsed; 2 stopped PD-1 therapy after 2 years of treatment; 2 did not provide reason for stopping PD -1 treatment; 1 treated with PD -1 for only 1 cycle due to economic reason. The other 2 were in the 100mg cohort: 1 stopped PD-1 therapy after 2+ years treatment and progressed 3m+ after the last PD-1 dose; 1 progressed 1 year after completing 2 years of PD-1 monotherapy. We have added the relative information above to Results section (Page 7, Line 139-140).

Discussion: What type of agent is CC-486 (line 316)?

Response: Thank you for the question. CC-486 is a hypomethylating agent. We have clarified the type of CC-486 (Page 12, Line 261).

Did some patients continue therapy past progression (as seen in Figure F)?

Response: Thank you for the question. There were 6 patients continued therapy past progression.

Reviewer #4

Song et al report on a phase 1 trial using Toripalimab, an anti-BTLA antibody in 71 patients. The trial is broken up into two groups with the first one evaluating safety with monotherapy and a second cohort with Toripalimab and an anti-PD1 antibody, tificemalimab. There are not a lot of Anti-BTLA drugs, so this study is important as this new drug is in testing and per article, additional studies. Based on the data listed here, the combination did seem to help, and this included some that failed checkpoint inhibitors in the past. The trial design/methodology makes sense and was appropriate. I don't see any flaws in data analysis.

Response: Thank you for your recognition of our work and positive comments.

The description of the trial was lacking a few details though. 1) It states no DLTs were found but the DLT criteria was not described. There were grade 3/4 hematologic AE's. The results listed appropriate AE's/TRAE's/IRAE's and with the other PD1 inhibitors, this seemed to be about the same.

Response: Thank you for the comments. We have described the DLT criteria in Methods section (Page 15, Line 333-337).

2) The trials seemed to heavily favor Hodgkin lymphoma's efficacy and highlights limited/no activity in NHL. The biomarkers described in the procedures section seems like they would test all disease types but they do not report per disease type and only report on CHL HVEM and PD-L1. This would also help in future studies or reproducibility.

Response: Thank you for the comments. It is true that all disease types with sample collected in combination cohort were tested for HVEM and PD-L1 expression. However, the results were not show in the manuscript due to the small sample size which could not lead to clear conclusion. The biomarker expression data of the 4 non-Hodgkin lymphoma patients are shown below:

Patient number	HVEM lymphoma score	PD-L1 TPS
09003	20	NA
03009	3	<1%
02006	NA	<1%
10001	NA	<1%

3) Description of other trial sites, number of other sites.

Response: Thank you for the suggestion. We have added the description of other trial sites, number of other sites to Results section (Page 6, Line 115-116).

4) Statistics. "The sample size for indication expansion was to be adjusted based on the efficacy and safety data obtained during the study," seems vague. Was this a 3x3 or boin design?

Response: Thank you for the question. The sample size for the dose-escalation stage in our study was based on a 3+3 design. Since our study was an early phase study, the sample size for indication expansion calculation was not based on a statistical hypothesis, and the sample size would be adjusted based on the efficacy and safety data obtained during the study. So, this was neither 3+3 nor boin design.

A point-by-point response to the reviewers' comments:

Reviewer #1:

The authors have improved the manuscript with these revisions. The following comments remain:

Comment #1 follow-up: Thank you for providing this table. Please provide these data for T effector cells and cytokines in the supplement and make a note in the manuscript that this was assessed.

Response: Thank you for the suggestions. We have provided the data for T effector cells and cytokines in the supplement Table S8 and made a note in the manuscript as “peripheral blood samples were collected from 31 patients in either the Part A or Part B groups before and 24h after administration, while no significant changes in effector cells related to efficacy were observed (Supplementary Table S8).” Please refer to Page 9, Line 182-185 in the revision.

Comment #2 follow-up: Thank you for providing this information. Please add the ORR for relapsed cHL patients to the paragraph that details refractory patient responses. Please also add this information to table S7.

Response: Thank you for the suggestions. We have added the response information for relapsed cHL patients to the revision (Page 10, Line 207-210) as “In addition, there were 5 cHL patients in the RP2D cohort relapsed (defined as progression after 3 months from the last dose of PD-1) to PD-(L)1 blockade, and 3 of them responded, with an ORR of 60% (Supplementary Figure S2D, Supplementary Table S9)”. And we also added this information to the original Table S7, which was Table S9 in the updated Supplementary file.

Comment #3 follow-up: Thank you for the additional drug hold and toxicity data. Please indicate what “all these events were resolved with concomitant treatments” means – ie clarify if this was managed with corticosteroids or something else. Please also add the reason for drug discontinuation to Figure 1 when this was due to an adverse event. Please report TRAEs (Table 2) as grade 1-2, grade 3-4, and total. This will help the reader understand the severity of these AEs. Please report Table S2 with the specific events and grades 1-2 vs 3-4 and total that occurred for each AE type. Please add to table S4 with the specific events graded 1-2 vs 3-4, and total events that occurred for event – it is important for understanding toxicity to also have the lower grade AE details. Please also indicate what events led to dose interruptions in the table.

Response: Thank you for the suggestions. We have indicated what “all these events were resolved with concomitant treatments” means as “corticosteroids (5 patients) and adrenocortical hormone (1 patient)” (Page 8, Line 163-164), and added the reasons for drug discontinuation to the note below Figure 1 (Page 25, Line 475-479). We have revised Table 2 to report TRAEs as grade 1-2, grade 3-4, and total (Page 24), and Table S2 with the specific events and grades 1-2 vs 3-4 and total that occurred for each AE type. We have also added the specific events graded 1-2 vs 3-4, and total events to Table S4, and indicated what events led to dose interruptions in Table S5.

Clarifying comments:

A) Figure 2B legend states: “One patient in Part A and 3 patients in Part B without

baseline tumor evaluation data were not included in the figure.” Can the authors clarify if these patients were included in any of the efficacy assessments mentioned in the manuscript? Please also amend the number of efficacy evaluable patients to the manuscript text, as the text lists N=46.

Response: Thank you for the suggestions. One patient in Part A and 3 patients in Part B without post-baseline tumor evaluation data were included in the Full Analysis Set and all the efficacy assessments except for Figure 2A and 2B. In order to avoid confusion, we clarified this in the footnote below Figure 2 as “In Figure A and B, 1 patient in Part A and 3 patients in Part B were not included in the figures due to lack of post-baseline tumor evaluation data.” (Page 28, Line 504-505).

B) Please add to the manuscript the number of patients who continued past progression as well as their clinical outcomes.

Response: Thank you for the suggestions. We have added to the revision the number of patients who continued past progression as well as their clinical outcomes as “There were 6 patients in the 200mg cohort and 2 in the 100 mg cohort who continued therapy past progression, and all developed persistent PD except 1 in the 200 mg cohort who experienced PR after progression” (Page 10, Line 213-215).

C) Please report the patient level data for all HVEM and PD-L1 staining, including both cHL and NHL in the supplement. I agree with reviewer #4 that histology also should be added even if small sample size.

Response: Thank you for the suggestions. We have reported the patient level data for all HVEM and PD-L1 staining, including both cHL and NHL in the supplement Table S10, and added relative description to the revision (Page 11, Line 235-239).

D) With regard to reviewer #4, comment 3, the authors need to add the sample size information to the methods and also clarify what motivated the total number of patients enrolled.

Response: Thank you for the suggestion. We have added the sample size information to the methods as “The sample size for the dose-escalation stage was based on a 3+3 design. Since the study was an early phase study, the sample size for indication expansion calculation was not based on a statistical hypothesis, and the sample size would be adjusted based on the efficacy and safety data obtained during the study” (Page 14, Line 305-308), and also clarified what motivated the total number of patients enrolled as “Due to the limited anti-tumor activities observed in the patients with peripheral T-cell lymphoma, follicular lymphoma and diffuse large B-cell lymphoma, the patient enrolled was terminated early for those subtypes, while preliminary anti-tumor activities were observed in the cHL patients, therefore, the study continued to enroll cHL patients as planned, thus, a total of 71 patients were enrolled as of the study completion” (Page 6, Line 107-112).

A point-by-point response to the reviewers' comments:

Reviewer #3:

The authors have adequately addressed all of the reviewer comments.

I have a few minor additional comments.

For Figure 1, the numbers in a few of the boxes don't quite add up. For Part A, for 3 mg/kg dose, the box states that treatment discontinued in 8 patients, but the reasons add up to 9. For the 200 mg dose, the box states treatment discontinued in 6 patients, but the reasons add up to 7 (total number of patients treated). In part B, for the box Tifcemalimab 200 mg + toripalimab 240 mg, the box states that 5 patients discontinued treatment but there are only 2 patient reasons given.

Response: Thank you for pointing this out. We apologize for the errors and sincerely appreciate your meticulous work. We have corrected the numbers in Figure 1 (Page 29).

Would clarify what is meant by lack of post tumor baseline evaluation data since these patients were otherwise evaluable for all other parameters (this refers to the 1 patient in Part A and 3 patients in part B who didn't have post baseline evaluation). Would state more explicitly why evaluable for response but not target lesion evaluation.

Response: Thank you for the suggestion. 1 patient in Part A and 3 patients in part B who didn't have post baseline evaluation were not presented in the figures about target lesion evaluation (this refers to the plots for showing the best change of target from baseline), while they were included in the figures for response evaluation (this refers to swimming pool plots for tifcemalimab monotherapy and in combination with toripalimab) and all other parameters, as they had baseline data and had received at least one dose of the study drug and were included in the FAS (full analysis set) per the protocol. In the protocol, FAS is designated as the primary analysis population for this study. We have clarified this in the figure legend (Page 34, Line 592-595).

Figure 2E was a little confusing to me. If a patient is still on treatment, why were there responses shown past the arrow? I also couldn't find the patient who had partial response after progressive disease in part B.

Response: Thank you for pointing this out. Due to our negligence, we posted the wrong picture in the article. We apologize for this mistake and sincerely appreciate the finding, and we have replaced it with the correct figure (please refer to Page 34, Figure 2B), relative description in the text has also been corrected (Page 12, Line 247-249). All the other figures and tables were double-checked and reviewed. No updates or corrections are needed for other data/figures/tables. From the updated Figure 2B, we can see 2 patients discontinued the treatment had response evaluations, this is because that they discontinued treatment due to intolerable therapeutic toxicity and had tumor evaluation records during follow-up. The patient (01020) in 200mg cohort had partial response after progressive disease.